# Real-Time Ultrasound-Computed Tomography Fusion with Volume Navigation to Assess Pancreatic Cystic Lesions

Manoj Mathew [1], Mayur Virarkar [2], Jia Sun [3], Khoan Thai [4], Mohammed Saleh [4], Manuel Menendez-Santos [2,*], Deepak Bedi [4], Jeffrey E. Lee [5], Matthew Katz [3], Vikas Kundra [4] and Priya Bhosale [4]

[1] Department of Radiology, The University of Texas Health Science Center at Houston, Houston, TX 77030, USA; manoj.mathew@pennmedicine.upenn.edu

[2] Department of Radiology, University of Florida College of Medicine, Jacksonville, FL 32209, USA; mayur.virarkar@jax.ufl.edu

[3] Department of Biostatistics, University of Texas MD Anderson Cancer Center, Houston, TX 77030, USA; jsun9@mdanderson.org (J.S.); mhgkatz@mdanderson.org (M.K.)

[4] Department of Radiology, University of Texas MD Anderson Cancer Center, Houston, TX 77030, USA; mosaleh@health.ucsd.edu (M.S.); dbedi@di.mdacc.tmc.edu (D.B.); vkundra@mdanderson.org (V.K.); priya.bhosale@mdanderson.org (P.B.)

[5] Department of Surgical Oncology, University of Texas MD Anderson Cancer Center, Houston, TX 77030, USA; jelee@notes.mdacc.tmc.edu

* Correspondence: manuel.menendez@jax.ufl.edu

**Abstract:** Transabdominal ultrasound is a promising imaging modality for pancreatic cystic lesions. This study aims to determine if transabdominal ultrasonography with CT fusion (TAUS-f) using volume navigation can be used to measure pancreatic cystic lesions (PCLs) compared to CT alone. We evaluated 33 patients prospectively with known PCLs. The readers evaluated each PCL's size and imaging characteristics on TAUS-f and CT alone. These were compared to endoscopic ultrasonography reports. A total of 43 PCLs from 32 patients were evaluated. The detection rate by TAUS-f was 93%. Two of the three undetected PCLs were in the tail of the pancreas. Inter-reader variabilities for TAUS-f and CT were 0.005 cm and 0.03 cm, respectively. Subgroup analysis by size and location demonstrated that inter-modality variability between TAUS-f and CT was smallest for lesions < 1.5 cm with a size difference of −0.13 cm for each reader and smallest in the pancreatic head with a size difference of −0.16 cm and −0.17 cm for readers 1 and 2. We found that TAUS-f effectively evaluates PCLs compared to CT alone, thus suggesting that it should be considered part of the surveillance algorithm for a subset of patients.

**Keywords:** pancreatic pncreatic cyst; transabdominal ultrasound; CT fusion; computed tomography; transabdominal ultrasonography with CT fusion; abdominal imaging

## 1. Introduction

The reported prevalence of pancreatic cystic lesions (PCLs) is up to 2.5% of the United States population, according to the Surveillance, Epidemiology, and End Results (SEER) Program [1]. Studies have shown incidental detection rates of 8% in the world population and a 12% pooled prevalence in the United States [2]. The rate of detection of PCLs has increased with more frequent use of cross-sectional imaging; reported detection rates range from 13.5% to 41.6% [3,4].

PCLs comprise a wide variety of lesions, ranging from benign to malignant. Some, such as serous cystadenoma, are benign with negligible risk of malignant transformation, while others are pre-malignant lesions. Lesions such as intraductal papillary mucinous neoplasms (IPMNs) have been reported to have malignant transformation rates of 11–30% for side branch IPMNs and 36–100% for main duct IPMNs [5]. At the same time, 14.9% of mucinous cystic neoplasms (MCNs) have been reported to have adenocarcinoma or high-grade dysplasia [6]. Patients with mucinous cystic lesions also have an increased

whole-gland risk of pancreatic ductal adenocarcinoma [7]. Imaging surveillance of PCLs is essential due to the risk of malignant transformation.

There are several published guidelines for the management of PCL based on imaging and clinical characteristics. Important imaging characteristics of PCLs include cyst location/morphology, cyst size, communication with the main pancreatic duct, the presence of "worrisome features" (cyst $\geq$ 3 cm, thickened/enhancing cyst wall, nonenhancing mural nodule, and main pancreatic duct caliber $\geq$ 7 mm), the presence of "high-risk stigmata" (obstructive jaundice, enhancing solid component within the cyst, and main pancreatic duct caliber $\geq$ 10 mm), and multiplicity [8]. The existing criteria recommend the use of computed tomography (CT), magnetic resonance imaging (MRI), and endoscopic ultrasonography (EUS) as the imaging modalities of choice for the evaluation of PCLs.

Transabdominal ultrasonography (TAUS) is frequently used for the initial evaluation of patients with abdominal pain and other abdominal pathologies. It offers high resolution for evaluating cystic structures and is commonly used in clinical practice to assess several intra-abdominal organs. Joen et al. suggested that using TAUS with correlative imaging such as CT, MRI, and EUS improves the detection of PCLs compared to TAUS alone [9]. Only a few studies of PCLs have used TAUS to evaluate the pancreas, owing to the difficulty in obtaining an appropriate acoustic window because of overlying bowel gas, body habitus, and the deep retroperitoneal location of the pancreas. Nonetheless, TAUS has already shown promise in the detection and follow-up of PCLs [9–11]. It is advantageous compared to MRI, CT, and EUS because of its noninvasive nature, widespread availability, and lower cost [12].

Additionally, recent studies have demonstrated the potential of deep learning techniques in medical imaging analysis. Praveen et al. employed ResNet-32 and FastAI to diagnose ductal carcinoma from 2D tissue slides, showcasing the applicability of deep learning in pathological image analysis [13]. Ullah et al. proposed a dual encoder-decoder framework for anatomical structure segmentation in chest X-ray images, underscoring the advancements in segmentation tasks through deep learning methodologies [14].

Our objective was to determine whether the utilization of volume navigation (VNav), which fuses previously acquired CT images to real-time TAUS, would provide a similar diagnostic capability to that of CT alone in the determination of the size and imaging characteristics. This leads to a decrease in radiation exposure and an alternative imaging modality for patients undergoing long-term surveillance. We aim to show that TAUS with CT fusion (TAUS-f) can provide a cost-effective alternative to CT alone for patients undergoing long-term surveillance.

The ability to measure and characterize PCLs with TAUS-f can provide an alternative imaging modality for evaluating pancreatic cysts, decreasing radiation exposure and cost for patients undergoing long-term surveillance.

## 2. Materials and Methods

### 2.1. Patients

We conducted a prospective study that received approval from the Institutional Review Board and complied with HIPAA regulations. Informed consent for participation in the study was obtained for 33 patients recruited from September 2012 through January 2017 at The University of Texas MD Anderson Cancer Center (MDACC). Eligibility criteria included patients with known PCLs who had undergone abdominal CT at MDACC.

### 2.2. Study Design

Following informed consent, each patient underwent TAUS-f within 30 days after the initial CT. The ultrasound exam was performed using a LOGIQ E9 US machine (GE Healthcare, Chicago, IL, USA) with volume navigation fusion imaging (VNav) to fuse CT images during the real-time ultrasound. VNav has the advantage of real-time fusion with the ability to scan in multiple planes while the CT image simultaneously rotates with the US. Before the TAUS-f started, DICOM (Digital Imaging and Communications in Media) axial

CT images were uploaded to the US machine and fused using VNav. A high-resolution 1–6 MHz curved-array transducer was used to evaluate the pancreas. VNav requires a clip on the transducer and transmitter by the bedside to accomplish the fusion. The PCLs were identified by the US technologist using TAUS-f. The US technologist and an abdominal radiologist identified the PCLs and labeled the location(s) of the lesion(s). The US images were saved as unfused and fused.

Of the 33 patients, 10 returned for an additional follow-up examination and underwent multiphase CT of the abdomen and pelvis followed by repeat TAUS-f. Patients who underwent cyst aspiration between the TAUS-f and CT were excluded. Figure 1 summarizes the overall study workflow.

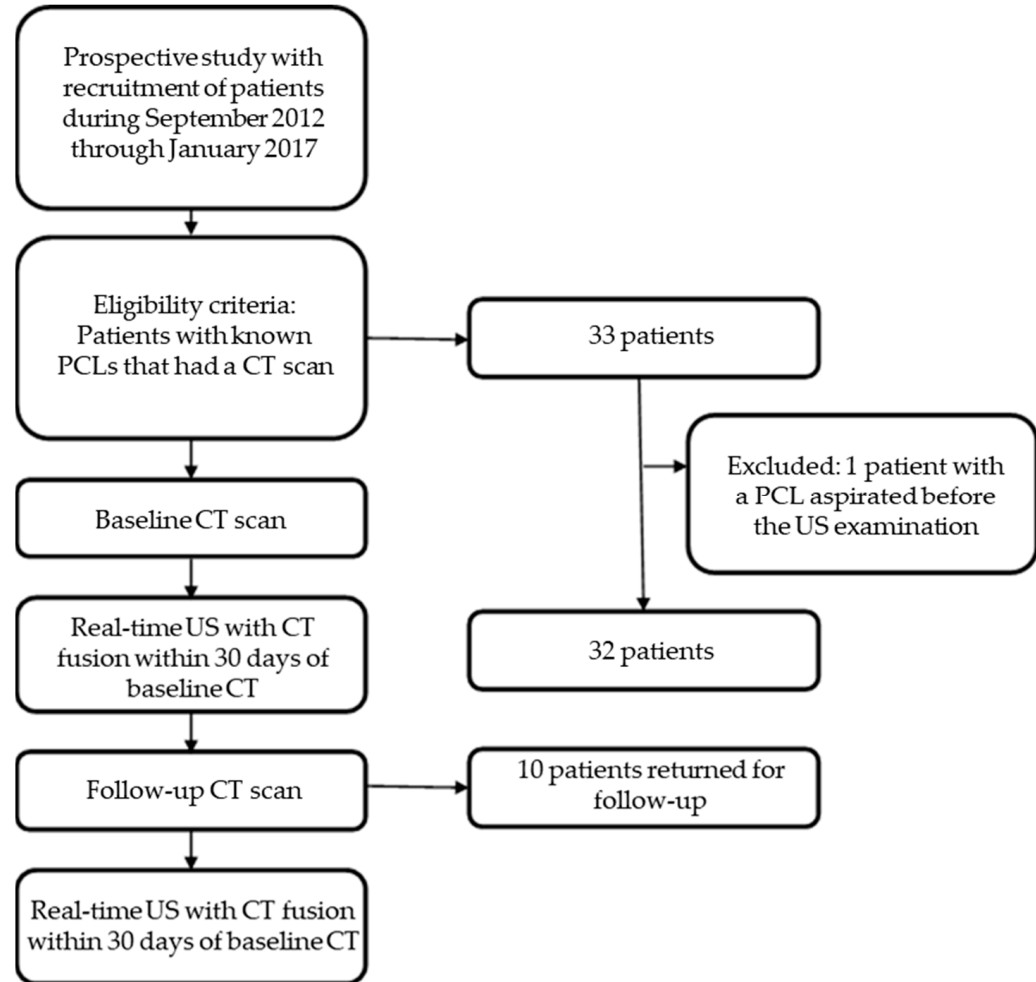

**Figure 1.** Flow chat shows the selection of patients and the overall study workflow. PCL, pancreatic cystic lesion; CT, computed tomography; US, ultrasonography.

### 2.3. Image Analysis

An attending abdominal radiologist with 20 years of experience and a radiology resident retrospectively and independently reviewed the images. First, the readers reviewed the unfused ultrasound images while blinded to the initial CT results to determine size and imaging characteristics (described below). Subsequently, the readers evaluated the CT images for the same features. The recorded PCL size was defined as the longest cross-sectional diameter in a single plane. The imaging characteristics evaluated were evidence of a soft-tissue component, main pancreatic duct dilation, and communication with the main pancreatic duct (MPD). The evidence of a soft-tissue component was defined as septations, internal echogenic or mass-like components, mural nodules, and wall thickening. In

addition, a chart review was completed retrospectively by the radiology resident to record the size and imaging characteristics of any EUS within 6 months of the TAUS.

### 2.4. Statistical Analysis

Inter-reader and inter-modality variabilities in measured size were calculated using a two one-sided *t*-test (TOST). Inter-reader variability was calculated by determining the mean difference between the two readers in the measured size of each PCL with TAUS-f and CT. Equivalence was then determined using the TOST with margins of 0.5 cm. Inter-modality variability was calculated by determining the mean difference between TAUS-f, CT, and EUS in the measured size of each PCL, then again using the TOST with margins of 0.5 cm to determine equivalence. Subgroup analyses of measured size differences were completed according to the location of the PCL (head, neck, body, or tail) and the size criteria set by American College of Radiology (ACR) guidelines: <1.5 cm, 1.5–2.5 cm, and >2.5 cm. Bland Altman plots were also used to compare inter-modality measurements for each reader.

The inter-reader and inter-modality variability in each imaging characteristic (detection of a soft-tissue component, main pancreatic duct dilation, and main pancreatic duct communication) was calculated using the kappa statistic. The kappa statistic results were then categorized as <0.20 as poor agreement, 0.20–0.40 as fair agreement, 0.41–0.60 as moderate agreement, 0.61–0.80 as substantial agreement, and 0.81–1.00 as perfect agreement.

## 3. Results

### 3.1. Study Population and PCL Characteristics

Thirty-three patients were recruited; one patient was excluded because of cyst aspiration before TAUS-f. The 32 patients included in the study had a median age of 67 years; all underwent evaluation with CT followed by TAUS-f. Seventeen patients were men, and 15 were women (Table 1). Ten of the 32 patients returned for a follow-up examination, at which point they again underwent TAUS-f. Thus, 43 PCLs were evaluated with TAUS-f and compared to CT scans. A chart review showed that EUS evaluated 13 PCLs within 6 months of the TAUS. Most of the lesions were in the head of the pancreas (17/43), followed by the tail (11/43). The mean sizes of the PCLs measured by readers 1 and 2 were 2.40 cm and 2.41 cm on TAUS and 2.45 cm and 2.48 cm on CT, respectively (Table 2).

**Table 1.** Characteristics of Study Patients and PCLs.

| Characteristic | Value |
|:---:|:---:|
| **Sex** | |
| Male | 17/32 |
| Female | 15/32 |
| **Age (y)** | |
| Median | 67 |
| Range | 46–85 |
| **Patients who presented for follow-up** | 10 |
| **Location of PCL (no. of lesions)** | |
| Head | 17/43 |
| Neck | 5/43 |
| Body | 10/43 |
| Tail | 11/43 |
| **PCLs detected by TAUS (no. of lesions)** | 40/43 |
| **Patients with EUS within 6 months of TAUS (no.)** | 13 |

PCL, pancreatic cystic lesion; TAUS, transabdominal ultrasonography; EUS, endoscopic ultrasonography.

**Table 2.** Measured size of detected PCLs by modality and reader.

|  | Reader 1 | Reader 2 |
|---|---|---|
| **Size of PCLs** | | |
| <1.5 cm | | |
| TAUS | 23 | 19 |
| CT | 22 | 21 |
| 1.5–2.5 cm | | |
| TAUS | 7 | 11 |
| CT | 7 | 10 |
| >2.5 cm | | |
| TAUS | 14 | 10 |
| CT | 10 | 11 |
| **Mean size of detected PCLs (cm)** | | |
| TAUS | $2.40 \pm 2.65$ | $2.41 \pm 2.56$ |
| CT | $2.45 \pm 2.43$ | $2.48 \pm 2.53$ |
| EUS * | | $2.73 \pm 2.10$ |

| **Intra-reader variability in measured size** | **Reader 1 vs. Reader 2** |
|---|---|
| TAUS | $-0.01 \pm 0.39$ |
| CT | $-0.03 \pm 0.43$ |

PCL, pancreatic cystic lesion; TAUS, transabdominal ultrasonography; CT computed tomography; EUS, endoscopic ultrasonography. * A single measurement obtained directly from the report of the EUS Equivalence test (two one-sided $t$-test) indicates significant evidence of equivalence ($p < 0.05$).

### 3.2. Detection of the Lesions

A total of 93% of the PCLs found on CT were also identified on TAUS-f (40/43). The three lesions that TAUS-f did not detect measured 1.0 cm; two lesions were in the tail, and one was in the head of the pancreas.

### 3.3. Inter-Reader Variability

The mean difference between readers in the measured size of PCLs was 0.005 cm on TAUS-f and 0.03 cm on CT (Table 2).

For the imaging features obtained on TAUS-f, there was moderate agreement between readers on the presence of a soft-tissue component and MPD dilation, with kappa statistics of 0.58 and 0.47, respectively (Table 3), which were statistically significant (<0.05). During the CT evaluation of imaging features, there was only fair agreement between readers on the detection of a soft-tissue component and MPD dilation.

**Table 3.** Inter-reader and inter-modality variability of imaging characteristics.

| Comparison | Soft-Tissue Component | MPD Dilation | MPD Communication |
|---|---|---|---|
| **Inter-reader variability** | | | |
| TAUS | 0.58 | 0.47 | 0.23 |
| CT | 0.23 | 0.39 | 0.58 |
| **Inter-modality variability** | | | |
| TAUS vs. CT | | | |
| Reader 1 | 0.89 | 0.89 | 0.85 |
| Reader 2 | 0.70 | 0.66 | 0.60 |
| TAUS vs. EUS | | | |
| Reader 1 | 0.44 | $-0.20$ | 0.17 |
| Reader 2 | 0.07 | 0.55 | 0 |
| CT vs. EUS | | | |
| Reader 1 | 0.62 | $-0.19$ | 0.27 |
| Reader 2 | 0.13 | 0.21 | $-0.10$ |

TAUS, transabdominal ultrasonography; CT, computed tomography; EUS, endoscopic ultrasonography; MPD, main pancreatic duct.

*3.4. Inter-Modality Variability*

The difference in the size of all PCLs showed that the lesions measured largest on CT, followed by TAUS-f, and then EUS. PCLs measured 0.16 cm and 0.19 cm smaller on TAUS-f than CT for readers 1 and 2, respectively (Table 4). Although TAUS-f measured larger than EUS or CT, the differences in the measurements were not statistically significant except for reader 1 in the comparison of CT and EUS.

**Table 4.** Subgroup analysis of differences in measured size.

|  | TAUS vs. CT | TAUS vs. EUS | CT vs. EUS |
| --- | --- | --- | --- |
| **All PCLs (cm)** | | | |
| Reader 1 | −0.16 ± 0.53 * | 0.13 ± 1.14 | 0.15 ± 0.65 * |
| Reader 2 | −0.19 ± 0.55 * | 0.18 ± 0.81 | 0.22 ± 0.90 |
| **By location (cm)** | | | |
| Head | | | |
| Reader 1 | −0.16 ± 0.62 * | 0.07 ± 0.49 * | 0.14 ± 0.37 * |
| Reader 2 | −0.17 ± 0.60 * | 0.20 ± 0.63 | 0.11 ± 0.46 |
| Neck | | | |
| Reader 1 | −0.04 ± 0.18 * | NA | NA |
| Reader 2 | −0.28 ± 0.57 | NA | NA |
| Body | | | |
| Reader 1 | −0.37 ± 0.37 | 0.65 ± 1.84 | 0.43 ± 1.00 |
| Reader 2 | −0.21 ± 0.59 | 0.48 ± 1.17 | 0.70 ± 1.48 |
| Tail | | | |
| Reader 1 | −0.37 ± 0.23 * | −0.75 ± 0.35 | −0.35 ± 0.64 |
| Reader 2 | −0.13 ± 0.38 * | −0.45 ± 0.07 | −0.35 ± 0.50 |
| **By size category (cm)** | | | |
| <1.5 cm | | | |
| Reader 1 | −0.13 ± 0.23 * | −0.38 ± 0.45 | −0.26 ± 0.34 |
| Reader 2 | −0.13 ± 0.38 * | −0.13 ± 0.30 * | −0.06 ± 0.39 * |
| 1.5–2.5 cm | | | |

PCL, pancreatic cystic lesion; TAUS, transabdominal ultrasonography; CT, computed tomography; EUS, endoscopic ultrasonography; The PCLs that were detected in the neck on TAUS and CT had not been evaluated with EUS within 6 months. * Equivalence test (two one-sided *t*-test) indicated significant evidence of equivalence ($p < 0.05$) between modalities.

Although the overall patient sample size is small, subgroup analyses by location reveal equivalence between CT and TAUS-f in the head and body of the pancreas, with differences of 0.16 cm and 0.17 cm in the head and 0.02 cm and 0.15 cm in the body for readers 1 and 2, respectively (Table 4). Measurements in the tail were not equivalent between CT and TAUS.

Subgroup analyses by size category revealed that size measurements were equivalent between CT and TAUS for PCLs of <1.5 cm, with an average difference of 0.13 cm for both readers. PCLs that measured ≥1.5 cm on CT and TAUS were not equivalent except for lesions > 2.5 cm obtained by reader 1, which measured 0.06 cm (Table 4) smaller by TAUS than CT.

On imaging characteristics, an agreement between TAUS and CT was substantial for reader 1, with kappa statistics of 0.89, 0.89, and 0.85. For reader 2, the agreement was substantial to moderate, with kappa statistics of 0.70, 0.66, and 0.60 in the detection of a soft-tissue component, MPD dilation, and MPD communication, respectively (Table 3).

## 4. Discussion

TAUS is not currently considered a standard modality for evaluating PCLs, at least partly because of challenges with visualization of the pancreas. Like the current gold standard of PCL evaluation (EUS), TAUS uses US waves, which can provide superior contrast resolution of cystic structures and spatial resolution compared to other imaging modalities. Based on our study, we believe that TAUS-f can be used for the surveillance of PCLs in addition to CT or MRI for smaller PCLs located in the pancreatic head.

In this study, we sought to evaluate the ability of TAUS-f to assess the size and imaging characteristics of PCLs. TAUS-f allowed us to detect most of the PCLs found on CT, with an overall 93% detection rate. Prior studies have evaluated the value of TAUS in the evaluation of PCLs, with some promising results [9–11,13]. The detection rates of PCLs in these studies were variable and showed associations with PCL location and size. Sun et al. found poor detection rates for PCLs that were <1.0 cm and located in the tail of the pancreas: 34.6% for 0.5–1 cm PCLs and 15.8% for <0.5 cm PCLs [10].

Additionally, lesions in the tail were detected only 18.3% of the time. While Sun et al. blinded themselves to the same-day MRI results, Jeon et al. leveraged the presence of prior imaging to improve detection rates [9]. They found that correlating TAUS with CT, EUS, or MRI improved overall detection from 49.2% to 86.7%. Like Jeon et al., we leveraged cross-sectional imaging with real-time CT fusion during TAUS examination and felt that this technology's use has led to a higher overall detection rate of 93% compared to prior studies. Studies continue to show the poor performance of TAUS in the detection of pancreatic tail lesions and higher detection rates for PCLs located outside the tail at 89.5% vs. 65.0% [11]. Jeon et al. showed that even correlative imaging has the lowest impact on improving detection rates for PCLs in the tail [9]. We also faced similar challenges with two undetected lesions in the pancreatic tail.

Our study showed minor absolute size differences between modalities, with the largest measured sizes on CT, followed by TAUS-f and EUS. In general, prior studies have found that CT and MRI measure PCLs larger than EUS [13,14], which was also the case in our study. Maimones et al. showed that CT measures PCLs 0.17 cm larger than EUS, as we observed (0.16 cm for reader 1 and 0.19 cm for reader 2). When comparing imaging to pathologic size, Maimones et al. also found that EUS tends to overestimate, while Huynh et al. found that EUS slightly underestimates. Given that TAUS-f also tends to measure lesions larger than EUS, TAUS-f likely overestimates the size of PCLs. The literature supports our findings of differences in inter-modality measurements, with cross-sectional imaging frequently overestimating the size of PCLs compared to the pathology [13,15]. The differences in size between CT and TAUS-f in comparison with EUS may have also been impacted by cyst aspiration at EUS, which was sometimes conducted before obtaining either the CT or TAUS-f.

Based on measurements by two readers with different experience levels, the subgroup analysis revealed that lesions < 1.5 cm in the pancreas' head or body were equivalent between TAUS-f and CT for both readers. This finding suggests that TAUS-f may be a helpful adjunct in the imaging surveillance algorithm for evaluating smaller PCLs and that even relatively inexperienced readers can obtain accurate measurements. In contrast, the assessment of larger lesions ($\geq$1.5 cm) showed inter-modality variability for both readers, with Bland-Altman plots showing a generally larger difference in measured size between modalities as the mean size of the lesion increases (Figure 2). Based on location, average measurements in the tail were smaller on TAUS-f compared to CT and EUS for both readers. Previous studies have demonstrated a similar finding with Aghdassi et al., showing that TAUS under-measured the size of PCLs compared to MRI, with the largest difference in the tail [16].

In evaluating imaging characteristics, there was substantial to moderate agreement between CT and TAUS in detecting a soft-tissue component, MPD dilation, and MPD communication. However, in comparing TAUS-f and CT to EUS, there was only moderate agreement in detecting a soft tissue component for reader 1. Du et al. also found that EUS was better than CT or MRI at detecting a soft-tissue component and septa [17]. Examples of soft tissue components demonstrated on CT, TAUS-f, and EUS are shown in Figures 3–5, with some cases also showing main pancreatic duct communication.

Our study has a few limitations. First, our sample size was small, and our patient population was limited to a single cancer institution. We also defined the presence of soft tissue components, which generally include septations, internal echogenic material, and a mural nodule. Clinically, mural nodularity is far more important, and additional

studies will need to be conducted to evaluate the ability of TAUS-f to specifically detect mural nodularity compared to cross-sectional imaging. Another limitation is that we had a small number of EUS exams for comparison and variability regarding when the EUS was performed for the TAUS-f. This evaluation would be necessary since EUS is the current gold standard for evaluation. Future studies will also be needed to evaluate the performance of TAUS-f in a subset of patients to determine the patients with optimal body habitus and limitations regarding hepatic steatosis, which has been found to make pancreatic visualization more challenging [18].

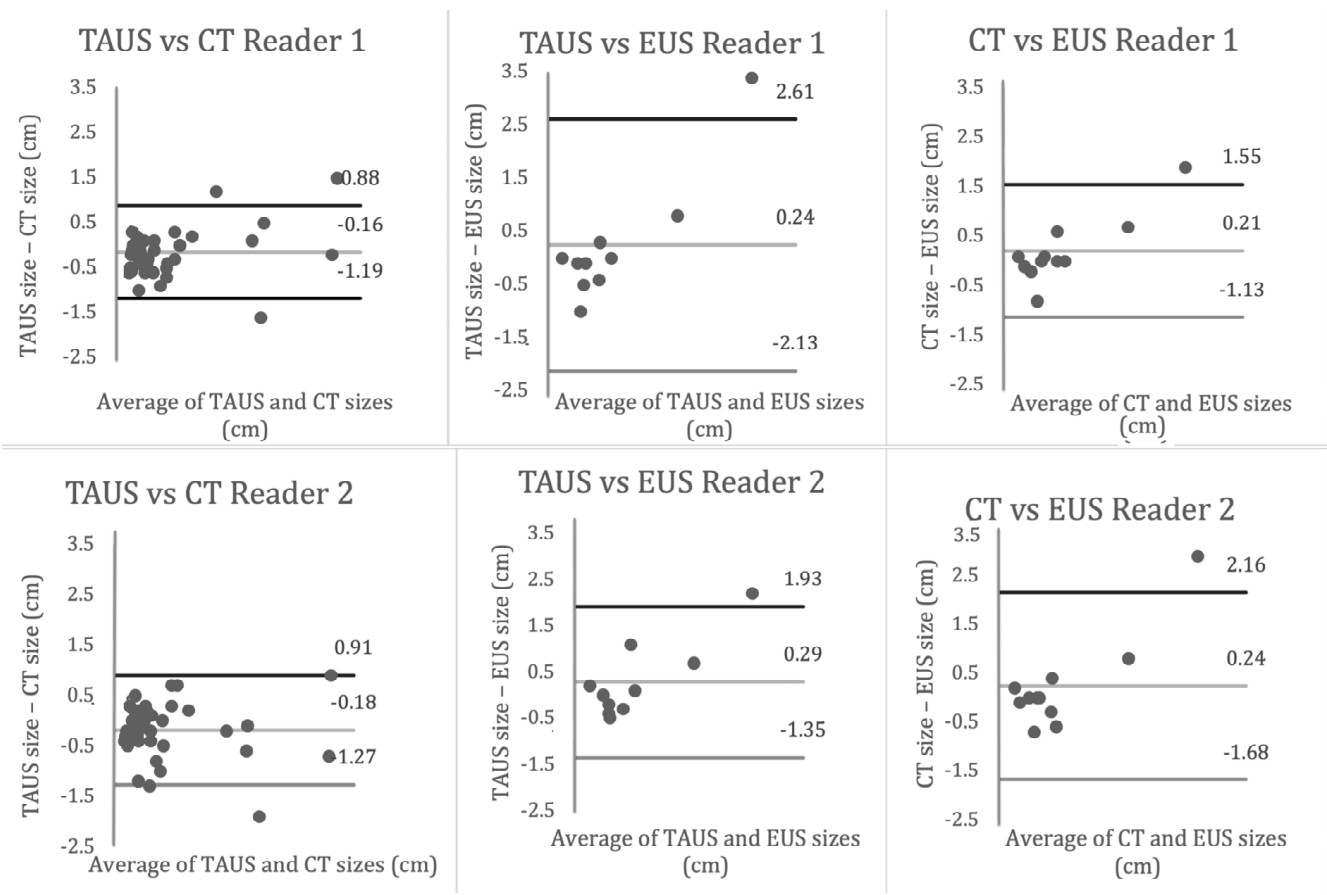

**Figure 2.** Bland Altman Curves.

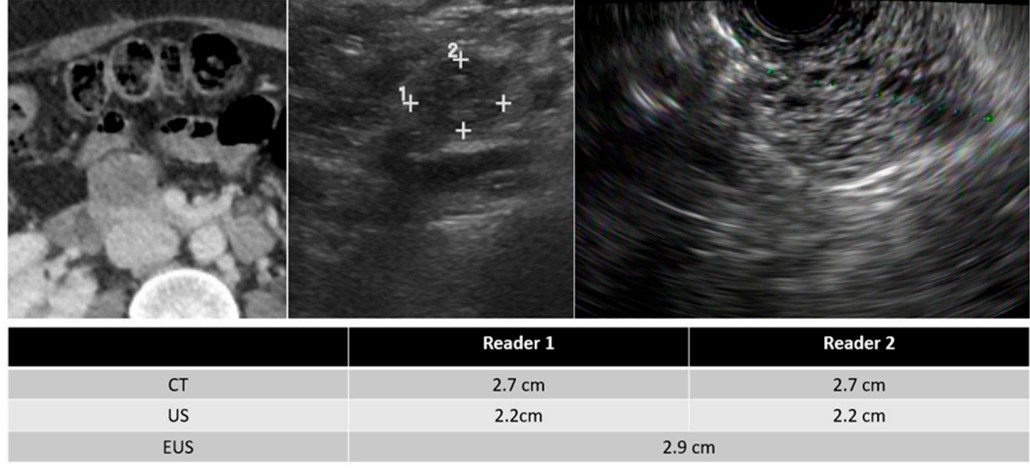

| | Reader 1 | Reader 2 |
|---|---|---|
| CT | 2.7 cm | 2.7 cm |
| US | 2.2cm | 2.2 cm |
| EUS | 2.9 cm | |

**Figure 3.** 55-year-old female with a serous cystadenoma and images of the CT (**left**), TAUS (**middle**), and EUS (**right**).

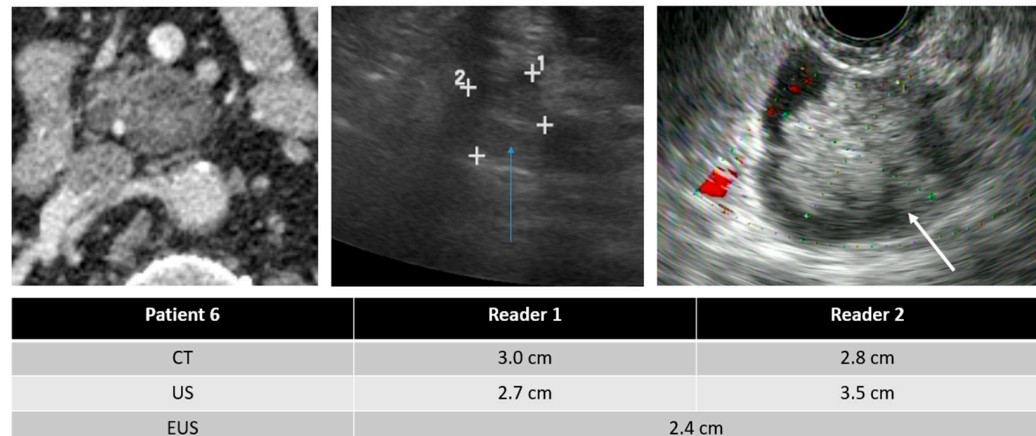

| Patient 6 | Reader 1 | Reader 2 |
|---|---|---|
| CT | 3.0 cm | 2.8 cm |
| US | 2.7 cm | 3.5 cm |
| EUS | 2.4 cm ||

**Figure 4.** 75-year-old male with a PCL and images from the CT (**left**), TAUS (**middle**), and EUS (**right**). EUS showed a 2.4cm cyst communicating with a focally dilated main pancreatic duct (6.5 mm). FNA is consistent with IPMN. Note the soft tissue nodule within the IPMN (white arrow), which is also seen on the TAUS (blue arrow).

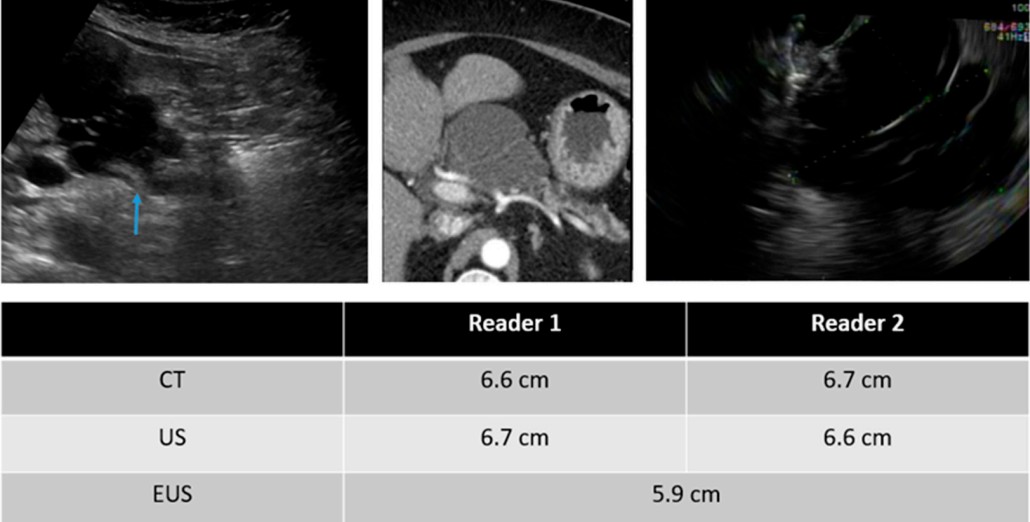

|  | Reader 1 | Reader 2 |
|---|---|---|
| CT | 6.6 cm | 6.7 cm |
| US | 6.7 cm | 6.6 cm |
| EUS | 5.9 cm ||

**Figure 5.** 78-year-old male with images of the TAUS (**left**), CT (**middle**), and EUS (**right**) with a multi-septated, multicystic lesion in the pancreatic head and body. Communication with the main pancreatic duct was visualized on TAUS (blue arrow). Surgical biopsy showed no viable malignant cells.

In conclusion, this study underscores the potential of transabdominal ultrasonography with CT fusion (TAUS-f) as a valuable adjunct in evaluating pancreatic cystic lesions (PCLs), offering novel insights and practical applications. Our research contributes to the ongoing discourse surrounding the optimal imaging modalities for PCL surveillance through a meticulous analysis of PCL detection rates, inter-reader and inter-modality variabilities, and the agreement of imaging characteristics.

This study's primary achievement lies in the demonstrated effectiveness of TAUS-f in detecting PCLs, achieving a notable 93% detection rate. This highlights TAUS-f's capability to enhance detection sensitivity and potentially identify PCLs that might have been missed through CT imaging alone. Moreover, identifying certain challenges in detecting lesions within the pancreatic tail sheds light on areas where further improvements or alternative approaches may be necessary.

By revealing minimal inter-reader variability in PCL size measurements using TAUS-f, we underscore the reproducibility and consistency of this novel method. This stability is

especially significant for clinicians and radiologists, as it signifies the potential to attain reliable results across different observers, even at varying experience levels. Our findings indicate that TAUS-f performs particularly well in assessing PCLs smaller than 1.5 cm, specifically within the pancreatic head. This insight directly impacts optimizing clinical surveillance algorithms for PCLs of different sizes and locations.

While this study showcases the promise of TAUS-f, we acknowledge the limitations associated with our sample size and the specific patient population studied. The computational complexity of TAUS-f warrants attention, and considerations regarding processing time, memory requirements, and clinical workflow impact are necessary for practical implementation.

Our research adds a critical layer of understanding to PCL evaluation, offering a fresh perspective on using TAUS-f and its potential implications for clinical practice. The insights garnered from this study provide a springboard for further investigations, necessitating larger cohorts, diverse patient demographics, and the exploration of optimization strategies. As medical technology advances and healthcare providers seek noninvasive and cost-effective imaging solutions, TAUS-f emerges as a promising contender with the potential to revolutionize how we approach the surveillance and management of pancreatic cystic lesions.

**Author Contributions:** All authors contributed to this paper with the conceptualization and design of the study, literature review and analysis, drafting, critical revision, editing, and final approval of the manuscript. All authors have read and agreed to the published version of the manuscript.

**Funding:** We thank The University of Texas MD Anderson Cancer Center for supporting this study, funded by an internal institutional grant. Grant number 2012-0110.

**Institutional Review Board Statement:** This study was conducted in accordance with the Declaration of Helsinki and approved by the Institutional Review Board of MD Anderson Cancer Center.

**Informed Consent Statement:** Informed consent was obtained from all subjects involved in this study.

**Data Availability Statement:** The data presented in this study are available on request from the corresponding author. The data are not publicly available to maintain patient confidentiality.

**Acknowledgments:** We thank our institution for supporting this study.

**Conflicts of Interest:** The authors declare no conflict of interest.

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
