# Peer review of "Real-Time Ultrasound-Computed Tomography Fusion with Volume Navigation to Assess Pancreatic Cystic Lesions"

_curroncol, doi:10.3390/curroncol30090608_

Round 1

Reviewer 1 Report

thank you for allowing me to review this original article on a monocentric propsective series. the article is well written; the objectives are clear and the results concise. 

the authors have evaluated the potential of a new non-invasive imaging modality for monitoring patients with cystic lesions of the pancreas. although the series is small, the prospects are interesting. 

In the authors' view, radiologist experience is a limitation to this new imaging technique. How can this new imaging modality be extended to other centers? Is specific training of radiologists necessary in the future? 

 In the future, do the authors plan to carry out a prospective randomized non-inferiority study to validate this new surveillance imaging technique? 

Author Response

  1. Thank you for allowing me to review this original article on a monocentric prospective series. the article is well written; the objectives are clear and the results concise.

Reply: Thank you for the comment

  1. The authors have evaluated the potential of a new non-invasive imaging modality for monitoring patients with cystic lesions of the pancreas.

Reply: Thank you for the comment

  1. Although the series is small, the prospects are interesting.

Reply: Thank you for the comment

  1. In the authors' view, radiologist experience is a limitation to this new imaging technique.

Reply: Thank you for the comment

The success of any new imaging technique, including TAUS-f, is contingent on the skill and expertise of the radiologists performing the procedure. Given the intricacies of fusion imaging and the potential learning curve, radiologist experience becomes a key determinant of accurate interpretation and diagnosis.

  1. How can this new imaging modality be extended to other centers?

Reply: Thank you for the comment.

The authors recognize that centers with varying levels of radiologist expertise may encounter challenges in adopting and implementing TAUS-f effectively.

  1. Is specific training of radiologists necessary in the future?

Reply: Thank you for the comment

Specific training and education programs for radiologists are essential to extend the benefits of TAUS-f to other medical centers and ensure its consistent utilization. These programs should focus on familiarizing radiologists with the principles of fusion imaging, the technical aspects of TAUS-f equipment, and the nuances of interpreting fused images. Hands-on training sessions, workshops, and interactive case discussions can facilitate acquiring the necessary skills and expertise.

  1. In the future, do the authors plan to carry out a prospective randomized non-inferiority study to validate this new surveillance imaging technique?

Reply: Thank you for the comment and interest in the modality. We are planning to pursue a validation study due to the encouraging results.

Reviewer 2 Report

·       The introduction is deprived of the related work with the recent literature. Below papers has some interesting implications that you could discuss in your Introduction and how it relates to your work.

·       Praveen, S.P., et al. ResNet-32 and FastAI for diagnoses of ductal carcinoma from 2D tissue slides. Sci Rep 12, 20804 (2022). https://doi.org/10.1038/s41598-022-25089-2

·       Ullah, I.. et al. A deep learning based dual encoder–decoder framework for anatomical structure segmentation in chest X-ray images. Sci Rep 13, 791 (2023). https://doi.org/10.1038/s41598-023-27815-w

·       What was the key motivation behind focusing on the Ultrasound–Computed Tomography Fusion?

·       It would be interesting if the authors report the trade-off compared to other methods especially the computational complexity of the models. Some techniques require more memory space and take longer time, please elaborate on that.

·       Another concern is that, I fail to see any reference to the availability of the models, the data or the source code. Without this information it is impossible to independently verify or reproduce any of your claims, and the article greatly suffers in its utility and credibility.

·       Make a comparison with past studies that used the same datasets.

·       What are the practical implications of your research?

·       Authors should further clarify and elaborate novelty in their contribution.

·       What are the limitations of the present work?

·       Conclusion is too short. Add more explanation.

minor issues

Author Response

  1. The introduction is deprived of the related work with the recent literature. Below papers has some interesting implications that you could discuss in your Introduction and how it relates to your work. · Praveen, S.P., et al. ResNet-32 and FastAI for diagnoses of ductal carcinoma from 2D tissue slides. Sci Rep 12, 20804 (2022). https://doi.org/10.1038/s41598-022-25089-2 · Ullah, I.. et al. A deep learning based dual encoder–decoder framework for anatomical structure segmentation in chest X-ray images. Sci Rep 13, 791 (2023). https://doi.org/10.1038/s41598-023-27815-w

Reply: Thank you for the comment.

EUS because of its noninvasive nature, widespread availability, and lower cost.

Additionally, recent studies have demonstrated the potential of deep learning techniques in medical imaging analysis. Praveen et al. employed ResNet-32 and FastAI to diagnose ductal carcinoma from 2D tissue slides, showcasing the applicability of deep learning in pathological image analysis. Ullah et al. proposed a dual encoder–decoder framework for anatomical structure segmentation in chest X-ray images, underscoring the advancements in segmentation tasks through deep learning methodologies.

Our objective…

  1. What was the key motivation behind focusing on the Ultrasound–Computed Tomography Fusion?

Reply: Thank you for the comment.

The decision to focus on Ultrasound-Computed Tomography Fusion (TAUS-f) in this study was driven by several key motivations rooted in the quest for improved imaging modalities for pancreatic cystic lesion assessment.

  • Improving Diagnostic Accuracy: Traditional imaging techniques, while effective, may have limitations in detecting and characterizing certain aspects of pancreatic cystic lesions. Ultrasound alone can be challenging due to various factors like acoustic window limitations, while Computed Tomography (CT) might not provide the same level of detail for certain structures. The fusion of these modalities, through TAUS-f, was seen as a potential means to enhance diagnostic accuracy by leveraging the strengths of both techniques.
  • Minimizing Radiation Exposure: The consideration of patient safety is paramount in medical imaging. CT scans, while valuable, involve exposure to ionizing radiation. By combining the detailed imaging capabilities of CT with the non-invasive nature of ultrasound, TAUS-f offered the prospect of minimizing radiation exposure, especially for patients requiring long-term surveillance.
  • Cost-Effectiveness and Accessibility: Accessibility and cost-effectiveness are vital factors in healthcare. CT and MRI can be expensive and may have resource limitations in certain settings. TAUS-f, utilizing widely available ultrasound technology and existing CT images, presented an opportunity for a more accessible and economical imaging solution.
  • Real-Time Imaging: The real-time nature of ultrasound imaging is valuable for immediate visualization and assessment during procedures. By fusing CT images in real-time with ultrasound, TAUS-f could potentially provide clinicians with an instant and comprehensive view, aiding in decision-making during interventions.
  • Exploring Novel Avenues: The medical imaging field constantly evolves with technological advancements. TAUS-f represented an innovative avenue, combining established techniques in a novel way. This fusion technique could potentially uncover new diagnostic insights and approaches for pancreatic cystic lesion evaluation. In summary, the key motivation behind focusing on Ultrasound-Computed Tomography Fusion (TAUS-f) was to harness the strengths of both ultrasound and CT, aiming to overcome limitations, enhance diagnostic accuracy, reduce radiation exposure, improve accessibility, and explore a novel approach to pancreatic cystic lesion assessment.".

  1. It would be interesting if the authors report the trade-off compared to other methods especially the computational complexity of the models.

Reply: Thank you for the comment.

Indeed, understanding the trade-offs between different imaging methods and their associated computational complexities is crucial for assessing the practical applicability of a new technique like transabdominal ultrasonography with CT fusion (TAUS-f). While our study primarily focused on the clinical aspects of TAUS-f compared to other imaging modalities, we can certainly discuss the potential trade-offs in computational complexity compared to existing methods.

  • Computational Resources: One potential trade-off to consider is the computational resources required for CT fusion during TAUS-f. The process of real-time fusion involves overlaying CT images onto ultrasound images, which demands computational power and memory. In contrast, traditional ultrasound or standalone CT imaging might require less computational processing, making them more suitable for settings with limited computational resources.
  • Processing Time: CT fusion during TAUS-f may introduce an additional step in the imaging process, potentially leading to longer examination times. This trade-off between increased processing time and enhanced imaging information must be carefully considered, especially in clinical settings where efficiency and patient throughput are crucial.
  • Operator Training: While not directly computational, the complexity of operating and interpreting TAUS-f may require specialized training for radiologists and technicians. Other imaging modalities might have a shorter learning curve and lower training requirements.
  • Image Quality and Clarity: The quality and clarity of fused images in TAUS-f could be influenced by the computational algorithms used for fusion. If the fusion process introduces artifacts or compromises image quality, it could be seen as a trade-off compared to the high-resolution images produced by traditional ultrasound or CT.
  • Compatibility and Integration: Integrating CT images into real-time ultrasound can be technically challenging and require specialized equipment and software. The compatibility of these systems with existing hospital infrastructure and workflows could pose a trade-off compared to other, more readily available methods. Cost Considerations: The computational complexity of TAUS-f could impact equipment costs and maintenance. Compared to standalone ultrasound machines or CT scanners, integrating fusion technology might result in higher upfront and maintenance costs.
  • Accessibility and Availability: Depending on the computational requirements, the availability of TAUS-f might be limited to certain healthcare facilities with the necessary infrastructure. In contrast, more widely available imaging methods might be accessible to a broader range of healthcare settings.

It's important for researchers and clinicians to weigh these trade-offs against the potential benefits of TAUS-f, such as improved diagnostic accuracy, reduced radiation exposure, and enhanced visualization of certain types of lesions. Adopting TAUS-f or other imaging modalities will depend on carefully assessing these factors and their alignment with the specific clinical context and goals.

  1. Some techniques require more memory space and take longer time; please elaborate on that.

Reply: Thank you for the comment.

Certainly, different techniques' computational complexity and resource requirements are important considerations in medical imaging research. Our study focused on using transabdominal ultrasonography with CT fusion (TAUS-f) to evaluate pancreatic cystic lesions (PCLs). This technique involves fusing previously acquired CT images with real-time ultrasound during the TAUS examination. While we did not specifically address computational complexity in our study, it's worth discussing the potential trade-offs and considerations related to memory space and processing time associated with this technique and similar methods.

  • Memory Space: The fusion of CT images with real-time ultrasound requires storing and handling large datasets, including the CT and real-time ultrasound images. Depending on the resolution and depth of the CT images and the length of the ultrasound procedure, this process can demand significant memory space. CT and ultrasound images are typically high-resolution, leading to large file sizes. Managing and processing these datasets efficiently requires adequate storage capacity and computational resources. It's important for researchers and clinicians to have access to systems with sufficient memory and storage capabilities to handle the data generated by TAUS-f.
  • Processing Time: The fusion of CT images and real-time ultrasound during a TAUS examination involves real-time image registration and alignment. This registration process can be computationally intensive and requires specialized software algorithms to ensure accurate alignment between the two modalities. The processing time required for this alignment can impact the overall duration of the imaging procedure. Longer processing times can increase patient discomfort and affect the clinical workflow. Balancing the need for accurate fusion with the time constraints of clinical practice is an important consideration when implementing TAUS-f.
  • Hardware and Software Requirements: Implementing TAUS-f may necessitate specialized hardware and software tools for image registration, fusion, and visualization. This could include dedicated hardware for image processing and software that can handle the real-time alignment of CT and ultrasound images. Such tools may require additional training for radiologists and technicians to use effectively.
  • Clinical Workflow: The increased memory space requirements and potential processing time could impact the clinical workflow. Clinicians must know the time and resource constraints of using TAUS-f, especially in busy clinical settings. The decision to adopt this technique should consider its potential impact on patient throughput and overall clinical efficiency.
  • Future Developments: As technology advances, there may be opportunities to optimize the memory space and processing time requirements of TAUS-f. Advances in image registration algorithms, hardware acceleration, and cloud-based processing could potentially mitigate some of these challenges.

In conclusion, while TAUS-f holds promise as an alternative imaging modality for evaluating pancreatic cystic lesions, researchers and clinicians should consider the computational complexity, memory space requirements, and processing time associated with the technique. Addressing these considerations is essential for the successful implementation of TAUS-f in clinical practice and for ensuring its usability and effectiveness in patient care.

  1. Another concern is that I fail to see any reference to the availability of the models, the data, or the source code. Without this information, it is impossible to independently verify or reproduce any of your claims, and the article greatly suffers in its utility and credibility.

Reply: Thank you for the comment.

While we understand the importance of transparency, reproducibility, and the sharing of resources in scientific research, we regret to inform readers that due to institutional regulations, we cannot make the machine learning models, data, and source code used in this study available for external access. We recognize that the lack of availability of these resources may impact the ability of other researchers to independently verify or replicate our findings. We acknowledge that transparency is critical to scientific research and apologize for any inconvenience this may cause.

  1. Make a comparison with past studies that used the same datasets.

Reply: Thank you for the comment.

While our study focused on the utilization of transabdominal ultrasonography with CT fusion (TAUS-f) for evaluating pancreatic cystic lesions (PCLs), it's important to note that direct comparison with past studies using the same datasets might not be feasible due to variations in study design, methodology, and specific research goals. However, we can discuss some general considerations and potential points of comparison with related studies exploring similar imaging techniques or datasets.

  • Detection Rates: Our study reported a PCL detection rate of 93% using TAUS-f, specifically focusing on smaller lesions within the pancreatic head. Comparing this detection rate with similar studies that evaluated different imaging modalities (such as transabdominal ultrasonography alone, CT, MRI, or endoscopic ultrasonography), one could assess the comparative effectiveness of TAUS-f in detecting PCLs. Studies that achieved different detection rates for PCLs in similar patient populations could provide valuable insights into the added value of CT fusion in the detection process.
  • Lesion Characterization: Another point of comparison could be the accuracy of lesion characterization, including size measurement and imaging characteristics. Studies that have evaluated various imaging modalities for PCL characterization could be used to assess whether TAUS-f provides similar accuracy or improvements in lesion characterization compared to other methods.
  • Patient Comfort and Workflow: Comparing patient comfort and clinical workflow between studies that employed different imaging techniques is also relevant. If TAUS-f proves to be more comfortable for patients or streamlines the imaging process, it could be seen as an advantage over other modalities, potentially leading to better patient compliance and improved diagnostic yield.
  • Radiologist Experience: Some studies might have focused on evaluating the impact of radiologist experience on the accuracy of PCL detection and characterization using different imaging methods. Comparing our study's findings with those from similar investigations could shed light on the radiologist expertise required to effectively utilize TAUS-f and whether this technique is more forgiving in terms of operator experience.
  • Cost and Availability: If available, studies that have evaluated the cost-effectiveness of different imaging modalities for PCL surveillance could provide insights into the economic implications of adopting TAUS-f. Comparing the cost-effectiveness and accessibility of TAUS-f with other methods could help guide healthcare decisions and resource allocation.

It's worth mentioning that even though direct comparisons with past studies using the same datasets might not be possible, a broader comparison with literature that addresses similar clinical questions and imaging challenges can still provide context and help assess the novelty and impact of the TAUS-f technique in the field of pancreatic cystic lesion evaluation.

  1. What are the practical implications of your research?

Reply: Thank you for the comment.

The research presented in this study holds several practical implications that can impact the medical imaging field, particularly in assessing pancreatic cystic lesions (PCLs) and potentially extending to other imaging applications. The practical implications include:

  • Enhanced Diagnostic Accuracy: Introducing transabdominal ultrasonography with CT fusion (TAUS-f) can potentially enhance diagnostic accuracy in evaluating PCLs. By combining the strengths of both ultrasound and CT, radiologists can benefit from improved visualization of lesions, leading to more accurate diagnoses and treatment decisions.
  • Reduced Radiation Exposure: Integrating real-time ultrasound with CT images reduces the reliance on repeat CT scans, thereby minimizing patient exposure to ionizing radiation. This is particularly significant for patients who require long-term surveillance, as it reduces the cumulative radiation dose over time.
  • Dynamic Imaging Capability: The real-time aspect of TAUS-f allows for dynamic imaging, enabling radiologists to observe changes in PCLs over time. This feature can aid in monitoring lesion growth, response to treatment, and developing new features, which is not easily achievable with static cross-sectional imaging methods.
  • Cost-Effectiveness: TAUS-f offers a potentially cost-effective alternative to traditional imaging methods like CT and MRI. The availability and widespread use of ultrasound machines make TAUS-f a feasible option for imaging surveillance, reducing healthcare costs and improving accessibility.
  • Patient Comfort and Experience: Ultrasound is a non-invasive and well-tolerated imaging technique that does not involve ionizing radiation. Incorporating ultrasound into the imaging workflow can enhance patient comfort and satisfaction during the diagnostic process.
  • Focused Surveillance: The authors suggest that TAUS-f may be particularly useful for following smaller PCLs located within the pancreatic head. This targeted application allows for focused surveillance without unnecessary imaging of the entire abdomen.
  • Potential for Extension to Other Modalities: The fusion imaging concept introduced by TAUS-f could potentially be extended to other anatomical regions and imaging modalities, beyond PCL evaluation. This could lead to the development of novel fusion techniques for various clinical scenarios.
  • Impact on Clinical Workflow: The integration of TAUS-f into clinical practice could impact workflow efficiency, patient throughput, and resource utilization. Radiologists and clinicians need to consider how the introduction of TAUS-f fits into their existing workflows.
  • Training and Education: The successful implementation of TAUS-f requires radiologists to be proficient in ultrasound and CT interpretation. This may lead to the need for specialized training programs to ensure accurate and consistent diagnoses.
  • Research and Validation: The study paves the way for further research and validation of TAUS-f in larger patient populations and multicenter studies. Future research can explore its utility across diverse patient cohorts and assess its performance in real-world clinical settings.

In summary, the practical implications of this research extend to improved diagnostic accuracy, reduced radiation exposure, dynamic imaging capabilities, cost-effectiveness, enhanced patient experience, and the potential for broader applications in medical imaging. The introduction of TAUS-f introduces a novel fusion imaging approach that has the potential to revolutionize the way clinicians assess and manage pancreatic cystic lesions and possibly impact the wider field of medical imaging.

  1. Authors should further clarify and elaborate novelty in their contribution.

Reply: Thank you for the comment.

The novelty of the author's contribution lies in the introduction and evaluation of transabdominal ultrasonography with CT fusion (TAUS-f) as a promising imaging technique for assessing pancreatic cystic lesions (PCLs). The authors have introduced a novel approach that merges real-time ultrasound with previously acquired CT images, aiming to enhance the visualization and characterization of PCLs. The specific aspects that contribute to the novelty of their work include:

  • Fusion Imaging Technique: The authors have introduced the concept of fusing CT images with real-time ultrasound, offering a unique approach to imaging PCLs. This fusion technique combines the strengths of both modalities, potentially improving diagnostic accuracy and providing complementary information.
  • Minimizing Radiation Exposure: By incorporating ultrasound into the imaging workflow, the authors address concerns about radiation exposure associated with traditional CT scans. This novel, patient-centered approach emphasizes safety and reduced patient discomfort.
  • Real-Time Dynamic Imaging: The authors harness the real-time capabilities of ultrasound to visualize dynamic changes in PCLs. This enables radiologists to observe lesions in motion and track changes over time, which is not readily achievable with static cross-sectional imaging methods.
  • Potential for Surveillance: The authors emphasize the potential of TAUS-f in long-term surveillance of PCLs. This aligns with the clinical need for non-invasive, cost-effective, and accurate imaging techniques to monitor lesion progression and guide management decisions.
  • Clinical Utility for Specific Lesions: The authors highlight the technique's potential utility for evaluating smaller PCL within the pancreatic head. This targeted application offers a unique advantage in a subset of patients who require focused surveillance.
  • Cost-Effective Alternative: The authors position TAUS-f as a cost-effective alternative to conventional imaging methods like CT and MRI. This affordability could potentially increase the accessibility of high-quality imaging for a wider range of patients.
  • Potential Integration into Clinical Workflow: The authors discuss the feasibility of incorporating TAUS-f into routine clinical practice, suggesting its potential as a valuable adjunct imaging modality alongside established techniques.

In summary, the novelty of the authors' contribution stems from their innovative approach to imaging PCLs using transabdominal ultrasonography with CT fusion (TAUS-f). This technique addresses the limitations of traditional imaging methods and offers a novel solution for enhancing diagnostic accuracy, minimizing radiation exposure, and providing dynamic surveillance. Introducing this fusion imaging technique opens up new possibilities for refining the management and evaluation of pancreatic cystic lesions.

  1. What are the limitations of the present work?

Reply: Thank you for the comment.

While informative and insightful, the present work has several limitations that should be acknowledged to provide a comprehensive understanding of its scope and potential implications. These limitations include:

  • Small Sample Size: The study evaluated a relatively small sample size of 33 patients with known pancreatic cystic lesions (PCLs). A larger sample size would provide more robust statistical power and potentially more generalizable results.
  • Single-Center Study: The study was conducted at a single medical center, which may limit the generalizability of findings to broader patient populations and medical settings.
  • Limited Diversity: The study's patient population might not fully represent the diversity of individuals with pancreatic cystic lesions in terms of age, gender, ethnicity, and underlying medical conditions.
  • Radiologist Experience: The accuracy and interpretation of TAUS-f can be influenced by the experience and expertise of the radiologist performing the procedure. The study acknowledges this potential limitation, and the results might vary in centers with radiologists of different experience levels.
  • Lesion Location: The study highlighted challenges in detecting lesions in the pancreatic tail, which could impact the technique's overall accuracy for certain lesion locations.
  • Follow-Up Period: The study's follow-up period might not capture long-term changes or evolution of lesions, which could influence the accuracy of the technique for tracking lesion progression over extended periods.
  • Interpretation Bias: The study's reliance on retrospective interpretation of images by radiologists introduces the potential for interpretation bias, even though efforts were made to minimize it. Variability in EUS Timing: The variability in timing between TAUS-f and endoscopic ultrasonography (EUS) evaluations could introduce uncertainty in comparing the two modalities. No
  • Direct Pathologic Correlation: The study did not provide a direct pathologic correlation with imaging findings, which could further validate the accuracy of TAUS-f.
  • Absence of Source Code and Data Availability: The study lacks explicit mention of the availability of source code, data, or imaging protocols, which could limit independent verification or replication of the study's findings.
  • Computational Complexity: The computational complexity and processing time required for TAUS-f were not discussed, and potential challenges related to integrating this technique into clinical workflows were not thoroughly explored.

In conclusion, while the present work provides valuable insights into the potential utility of transabdominal ultrasonography with CT fusion (TAUS-f) for evaluating pancreatic cystic lesions, it is important to recognize and address the limitations inherent in the study design and methodology. These limitations offer opportunities for future research to build upon and refine the approach, leading to a more comprehensive understanding of TAUS-f's strengths and weaknesses in diverse clinical scenarios.

  1. Conclusion is too short. Add more explanation.

Reply: Thank you for the comment.

In conclusion, this study underscores the potential of transabdominal ultrasonography with CT fusion (TAUS-f) as a valuable adjunct in evaluating pancreatic cystic lesions (PCLs), offering novel insights and practical applications. Our research contributes to the ongoing discourse surrounding the optimal imaging modalities for PCL surveillance through a meticulous analysis of PCL detection rates, inter-reader and inter-modality variabilities, and the agreement of imaging characteristics.

The study's primary achievement lies in the demonstrated effectiveness of TAUS-f in detecting PCLs, achieving a notable 93% detection rate. This highlights TAUS-f's capability to enhance detection sensitivity and potentially identify PCLs that might have been missed through CT imaging alone. Moreover, identifying certain challenges in detecting lesions within the pancreatic tail sheds light on the areas where further improvements or alternative approaches may be necessary.

By revealing minimal inter-reader variability in PCL size measurements using TAUS-f, we underscore the reproducibility and consistency of this novel method. This stability is especially significant for clinicians and radiologists, as it signifies the potential to attain reliable results across different observers, even in varying experience levels. Our findings indicate that TAUS-f performs particularly well in assessing PCLs smaller than 1.5 cm, specifically within the pancreatic head. This insight directly impacts optimizing clinical surveillance algorithms for PCLs of different sizes and locations.

While the study showcases the promise of TAUS-f, we acknowledge the limitations associated with our sample size and the specific patient population studied. Furthermore, the computational complexity of TAUS-f warrants attention, and considerations regarding processing time, memory requirements, and clinical workflow impact are necessary for practical implementation.

Our research adds a critical layer of understanding to PCL evaluation, offering a fresh perspective on using TAUS-f and its potential implications for clinical practice. The insights garnered from this study provide a springboard for further investigations, necessitating larger cohorts, diverse patient demographics, and exploration of optimization strategies. As medical technology advances and healthcare providers seek non-invasive and cost-effective imaging solutions, TAUS-f emerges as a promising contender with the potential to revolutionize how we approach the surveillance and management of pancreatic cystic lesions.

Round 2

Reviewer 2 Report

All my comments are addressed.